# Does Age Influence the Preoperative Condition and, Thus, the Outcome in Endocarditis Patients?

**DOI:** 10.3390/jcm12030822

**Published:** 2023-01-19

**Authors:** Roya Ostovar, Farzaneh Seifi Zinab, Filip Schröter, Martin Hartrumpf, Dirk Fritzsche, Johannes Maximilian Albes

**Affiliations:** 1Department of Cardiovascular Surgery, Heart Center Brandenburg, University Hospital Brandenburg Medical School, Faculty of Health Sciences Brandenburg, 16321 Bernau, Germany; 2Department of Cardiac Surgery, Sana-Heart Center Cottbus, 03048 Cottbus, Germany

**Keywords:** outcome, aging, cardiac surgery, infective endocarditis, valve surgery

## Abstract

**Background:** Demographic changes have led to an increase in the proportion of older patients undergoing heart surgery. The number of endocarditis cases is also steadily increasing. Given the sharp increase in patients who have received valve prostheses or electrophysiological implants, who are on chronic dialysis or taking immunosuppressants, the interdependence of these two developments is quite obvious. We have studied the situation of older patients suffering from endocarditis compared to younger ones. Are they more susceptible, and are there differences in outcomes? **Patients and Methods:** A total of 162 patients was studied from our database, enrolled from 2020 to 2022. Fifty-four of them were older than 75 years of age (mean age 79.9 ± 3.8 years). The remaining 108 patients had a mean age of 61.6 ± 10.1 years. EuroSCORE II (ES II) was higher in the older patients (19.3 ± 19.7) than in the younger ones (13.2 ± 16.84). The BMI was almost identical. The preoperative NYHA proportions did not differ. A statistical analysis was performed using R. **Results:** Older patients had a lower left ventricular ejection fraction (LVEF), a higher proportion of coronary heart disease (CHD), a higher amount of N-terminal probrain natriuretic peptides (NT-proBNPs), worse coagulation function, worse renal function than younger patients, and were more often valve prosthesis carriers compared to the younger patients. The diagnostic interval was 66.85 ± 49.53 days in the younger cohort, whereas it was only 50.98 ± 30.55 in the elderly (*p* = 0.081). Significantly fewer septic emboli were observed in the older patients than in the younger patients, but postoperative delirium and critical illness polyneuropathy and critical illness myopathy (CIP/CIM) were observed significantly more frequently compared to younger patients. In-hospital mortality was higher in older patients than in younger patients, but did not reach statistical significance (29.91% vs. 40.38%; *p* = 0.256). The postoperative clinical status was worse in older patients than in the younger ones (NYHA-stage, *p* = 0.022). **Conclusions:** Age did have an impact on the outcome, probably due to causing a higher number of cumulative preoperative risk factors. However, an interesting phenomenon was that older patients had fewer septic emboli than younger patients. It can only be speculated whether this was due to a shorter diagnostic interval or lower mobility, i.e., physical exertion. Older patients suffered more frequently than younger ones from typical age-related postoperative complications, such as delirium and CIP/CIM. In-hospital mortality was high, but not significantly higher compared to the younger patients. Considering the acceptable mortality risks, and in light of the lack of alternatives, older patients should not be denied surgery. However, individual consideration is necessary.

## 1. Introduction

Infective endocarditis is regarded as one of the most threatening cardiac diseases in modern times, due to higher mortality and morbidity [1,2,3]. This disease seems to mainly affect the elderly [2,4,5,6]. With advancing age, not only does the prevalence of cardiac diseases increase but also extracardiac disease, which, directly or indirectly, due to medical interventions, can lead to an infection and, consequently, to endocarditis.

The elderly are at risk as a result of various causes. Diverse reasons contribute to this, including a suppressed immune system, cancer, more frequent treatments as outpatients or inpatients, dialysis dependency, and being valve prosthesis carriers. Furthermore, interventions and surgical procedures are additional reasons that can lead to infection [7]. A weakened immune system and altered metabolism in the elderly complicate not only their own intrinsic disease defense, but also therapeutic measures.

With a steadily aging society, a further increase in this disease is expected [8,9]. Although the aging of society is ubiquitous, especially in the medical field, we are apparently not best equipped to deal with this particularly vulnerable subgroup.

Although the indication for cardiac surgery in elderly patients needs to be considered depending on their risk profile, in some cases, it is unavoidable.

It is essential to understand how the preoperative condition of the elderly differs from the young. Furthermore, it is relevant to know the typical common courses and complications in the elderly, in order to address them appropriately. The aim of this study is, mainly, to analyze the outcomes of endocarditis patients after cardiac surgery in association with age.

## 2. Patients and Methods

Prior to the trial, approval was obtained from the responsible ethics committee (E-01-20191007, date: 20 January 2020). Written informed consent was obtained from the patients.

In this prospective multicenter study, data were collected from 162 patients between 2020 and 2022. Patients were divided into 2 age groups. A total of 54 patients ≥75 years and 108 patients <75 years was included in the study. The primary end point was to analyze the influence of age on the outcomes of patients after endocarditis. The secondary end point was to evaluate the effect of age on the preoperative condition.

Inclusion criteria were proven infective endocarditis according to the Duke criteria [10]. The exclusion criteria were patients under 18 years of age, patients who were treated conservatively and did not receive surgery, and patients who did not consent.

Concomitant diseases, risk factors, surgical or conservative therapeutic procedures, and postoperative courses and complications were collected. Moreover, perioperative diagnostics, such as echocardiography, electrocardiogram, and laboratory findings, including microbiological findings, were recorded.

A statistical analysis was performed using “R” (R Core Team, Vienna, Austria) [11]. Numerical data were tested for normal distribution using the Shapiro–Wilk test and compared between both age groups using the *t*-test and the Mann–Whitney U test, respectively. Categorical variables were analyzed using Fisher’s exact and Chi-squared tests. The ordinal variables were compared with the Cochran–Armitage test for trends in proportions.

## 3. Results

### Baseline, Risk Profile, and Comorbidities

EuroSCORE II was significantly higher in elderly patients (19.3% ± 19.7 vs. 13.2% ± 16.8%, *p* < 0.001). No significant differences were found in the gender, body mass index, and preoperative NYHA stage.

The proportion of patients with a prosthetic heart valve was significantly higher in older patients (54.7% vs. 35.2%, *p* = 0.028). Moreover, the proportion of patients with coronary artery disease and a reduced left ventricular ejection fraction was significantly higher in the elderly (*p* = 0.05; *p* = 0.017).

The proportion of patients with recurrent endocarditis, patients with early endocarditis (within 12 months after prosthesis implantation), proportion of redo surgeries, electrophysiology devices such as pacemakers or defibrillators, patients with carcinoma or having received chemotherapy, intra-abdominal and urogenital infections, and dental treatments showed no significant difference between the age groups, although the above items were more frequent in the elderly. In contrast, the proportion of patients with autoimmune diseases, patients under glucocorticoid treatments, with skin and soft tissue infections, bone and joint infections, respiratory tract infections, and drug and alcohol abuse was seen more frequently in younger patients, although the difference was not significant. Dialysis dependency and diabetes mellitus were comparable in both groups (Table 1).

## 4. Preoperative Diagnosis and Condition

The proportion of involved valves was comparable in both groups. The tricuspid valve was slightly more frequently affected in younger patients, without reaching significant levels. Moreover, no significant difference was found between the groups for germs detected in blood cultures or on the valves.

Septic embolism was significantly higher in younger patients (42.6% vs. 20.4%, *p* = 0.009). There was also a significant difference in the location of septic emboli (*p* = 0.019). While older patients were more often diagnosed with cerebral emboli, younger patients were more often diagnosed with septic emboli in the kidney, spleen, lung, and liver.

The diagnostic time frame from symptom onset to endocarditis diagnosis was, on average, 16 days longer in younger patients.

Heart failure, systemic inflammatory response syndrome, acute renal failure, delirium, cardiac arrhythmias, and pleural effusion were seen more frequently preoperatively in elderly patients, without reaching significant levels (Table 2).

The retention parameters, such as the expression of renal function, were significantly higher preoperatively in elderly patients (*p* < 0.001). Coagulation parameters were also significantly worse in elderly patients (*p* = 0.007). The number of N-terminal probrain natriuretic peptides (NT-proBNPs), as an expression of heart failure, was also significantly higher preoperatively in elderly patients (*p* = 0.002). Without reaching a level of significance, the elderly patients showed more anemia, a higher prevalence for inflammatory parameters, as well as elevated hepatic dysfunction parameters (Table 2).

## 5. Postoperative Course and Complications

In the postoperative course, the elderly patients developed significantly more delirium (42.5% vs. 20%, *p* = 0.013) and more critical illness myopathy and critical illness polyneuropathy (20.5% vs. 4.2%, *p* = 0.005). Without reaching a level of significance, more instances of postoperative bleeding, pleural effusion, and low-cardiac output syndrome were observed in the elderly patients. The duration of hospitalization was longer in the elderly patients (Table 3). The laboratory parameters were comparable postoperatively. The postoperative NYHA stage was significantly higher in the older patients (*p* = 0.022). In-hospital mortality was markedly higher in the elderly patients, without reaching a significant value (40.4% vs. 29.9%) (Table 3).

## 6. Discussion

Age has long been identified as one of the most important risk factors in cardiac surgery, probably even THE most important one, considering that, in current risk scores used for early outcome prediction, age strongly influences the score itself, especially for patients over 70 years [12]. The EuroSCORE assigns four points for patients over 75 years and five points for patients over 80 years. Five points alone resulted in a β-value of 1.67 for the logistic EuroSCORE and 0.68 for the EuroSCORE II, respectively, suggesting that advanced age is a significant risk indicator for premature mortality and is statistically proven. However, it is not clear what exactly causes advanced age to be such a significant risk factor. In a recent study, we found that in such an older cohort, renal insufficiency and its sequelae, such as postoperative renal failure, the need for dialysis, and, possibly, the development of pleural and pericardial effusions, negatively affected the outcome. In addition, we saw CIP/CIM more frequently in the most advanced age group, with a concomitant decrease in the BMI, indicating sarcopenia and, thus, an additional feature of frailty besides renal failure alone. Thus, the frail patients with renal failure, reduced muscle strength, and an already reduced neurocognitive abilities are at risk; therefore, it is certain that frailty and old age go hand in hand [13]. The topic of “frailty” is becoming increasing important in cardiac surgery. However, there is still no standardized, quick, and simple method for testing for frailty in the daily clinical practice in cardiac surgery. Thus, even in our patient cohort, different tests, such as hand pressure, the 5 min walk test, mobility, comorbidities, corresponding symptoms, and limitations, and mental tests, such as the mini mental state test, were performed. Due to this heterogeneity, frailty did not find a place in the analysis. However, we believe that frailty must be evaluated as an adjunct to the somatic EuroSCORE, as the age and comorbidities of our patients are steadily increasing. Currently, there is a steady increase in the proportion of patients of advanced age undergoing cardiac surgery [14,15]. The same applies to cardiac interventions, although corresponding register data are lacking. At the same time, the number of endocarditis cases is also steadily increasing [3]. It is more than obvious that these developments are interlinked. In view of the significant increase in patients who have received electrophysiological implants or valve prostheses, who are chronically dialysis-dependent, or who take immunosuppressants, an increased susceptibility to infections due to devices and native valves is to be expected. Consequently, we sought to identify specific age-related conditions or circumstances that favor the development of endocarditis, as well as specific problems in the perioperative course, in order to elicit potential targets for interventions to prevent or ameliorate them.

Furthermore, although it was not the focus of this study, there are other important factors that contribute to the development of endocarditis at an advanced age. The increasing number of pacemaker implantations, especially with advancing age, in recent decades, has resulted in an increased risk of endocarditis [16].

Another important risk factor in the development of endocarditis is dental and dental root infections. In our overall cohort, 14.35% of patients described symptom onset immediately after profound dental treatments. Presumably, hidden root canal infections and other nonprofound dental treatments play a role in the development of endocarditis, even if we do not yet have evidence for it.

Our study indeed showed that age has an influence. However, to our surprise, the older patients were found to have fewer septic embolisms than the younger patients. When looking at the diagnostic interval, it was found that the younger patients had a longer interval compared to the older patients. Although this was not statistically significant, it could still be speculated that a shorter diagnostic interval led to a lower likelihood of septic embolism due to growing vegetation. Another speculation could be that older patients are generally less mobile or less likely to undergo physical exertion, reducing the risk of mobilizing the otherwise adherent vegetations. Finally, it can be suspected that younger patients simply neglect the symptoms because they are absorbed in their respective professional activities.

Regarding the prerequisites, i.e., the preoperative conditions, renal function was significantly more often impaired in the older patients than in the younger cohort. The same was true for coagulation disorders and cardiac insufficiency, indicated by increased NT-proBNP levels. Finally, older patients were more likely to have anemia, higher inflammatory parameters, and increased parameters for hepatic dysfunction.

In the postoperative course, the older patients were found to suffer from delirium and CIP/CIM more frequently than the younger patients, indicating an already preoperatively altered neurocognitive functional state. In-hospital mortality was high in both cohorts, regardless of age. A high mortality rate in endocarditis also corresponded to the results of other studies [1,3,17,18,19,20]. However, it was not significantly higher in the older patients than in the younger ones in our cohort.

Furthermore, it should be noted that only surgically treated patients were included in this study. The patients who were not suitable for surgery because of a very high age, serious comorbidities, and severe frailty with very high risks of surgery and mortality were excluded. If every patient had undergone surgery regardless of age and frailty, a significant difference in mortality could likely be observed.

According to current data, approximately 5% to 10% of the aging population suffers from ischemic dementia [21]. Although ischemic dementia was not the subject of the present study, it is imaginable that a similar proportion of our patients may have suffered from it. It could be speculated that the patients with ischemic dementia might have had a worse outcome after surgery. However, this was not investigated in the present study, and further studies are needed to investigate this topic.

What can be deduced from these results? Firstly, shortening the diagnostic time frame is of utmost importance in all endocarditis patients. Secondly, elderly patients should be rapidly optimized as soon as an indication for surgery is given. This includes the correction of anemia and coagulation, the optimization of renal function by maintaining at least adequate diuresis, the assessment of neurocognitive abilities, and at least adequate psychological support, as well as thorough education about the disease and informed consent involving relatives to maintain a stable psychosocial state for the time being.

Postoperatively, the best available treatment, monitoring, and care are required to reduce the surgical burden and guide the elderly and frail patients through the early perioperative period. Awareness is key. Elderly and, thus, frail patients, do not forgive negligence.

On the one hand, elderly patients showed a high heterogeneity regarding their frailty and comorbidities. On the other hand, infective endocarditis showed a very high variability in outcomes depending on pathogen aggressiveness, the diagnostic time frame, protracted infection, and many other factors; thus, especially in elderly patients, there is no “one size fits all” treatment.

A breakthrough tool is not yet available, but the sum of all measures can be the difference needed to achieve an acceptable outcome.

## 7. Limitations

Given the small number of patients included in this analysis, the significance of the results is limited. However, it can be assumed that the registry’s database, which includes several hospitals, already provides a decent overview of the problem, which could become even more comprehensive in the future. The more patients are included and the broader the participation in the registry becomes, the more robust and valid the findings are likely to become. A future larger dataset might also be sufficient to study the effects of those baseline characteristics that were different between age groups without reaching significance.

## 8. Conclusions

In summary, as expected, the elderly patients were significantly more at risk. They were sicker from a cardiac standpoint, with significantly more CHD patients, more prosthesis carriers, worse left ventricular function, and higher NT-proBNP levels. Moreover, the elderly patients suffered more often from chronic diseases, including significantly more kidney failure. While in-hospital mortality was higher in the elderly, the difference was not significant. Considering the acceptable mortality risks and the lack of alternatives, surgery should not be denied to elderly patients. However, the challenges of infective endocarditis therapy always require an individual assessment, consideration, and planning.

## Figures and Tables

**Table 1 jcm-12-00822-t001:** Baseline, risk profile, and comorbidities.

Preoperative Condition	Patients < 75 Years	Patients ≥ 75 Years	*p*-Value
Gender (female)	23.2% [16.2–31.9]	29.6% [19.1–42.8]	0.482
EuroSCORE II (%)	13.2% ±16.8%	19.3 ± 19.7%	**<0.001**
Left ventricular ejection fraction	52.2% ± 10.2%	48.5% ± 10.6%	**0.017**
Body mass index (kg/m^2^)	28.9 ± 6.6	27.1 ± 3.9	0.113
Early endocarditis (%)	11.1% [6.5–18.4]	16.7% [9.0–28.7]	0.457
Recurrent endocarditis	5.6% [2.6–11.6]	11.1% [5.2–22.2]	0.34
Coronary heart disease	37.9% [29.4–47.4]	55.6% [42.4–68]	**0.05**
Redo surgery	32.4% [24.3–41.4]	38.9% [27.0–52.2]	0.521
Valve prosthesis carrier	35.2% [26.8–44.6]	54.7% [41.5–67.3]	**0.028**
Electrophysiology devices	26.9% [19.4–35.9]	35.2% [23.8–48.5]	0.362
Proportion of carcinoma	14.8% [9.3–22.7]	18.5% [10.4–30.8]	0.705
Proportion of chemotherapy	6.5% [3.2–12.8]	9.3% [4.0–19.9]	0.75
Glucocorticoid treatments	14.8% [9.3–22.7]	9.3% [4–19.9]	0.457
Autoimmune diseases	9.3% [5.1–16.2]	7.4% [2.9–17.6]	0.776
Diabetes mellitus	33.3% [25.2–42.7]	31.5% [20.7–44.7]	0.953
Dialysis dependency	5.6% [2.6–11.6]	5.6% [1.9–15.1]	1
Respiratory tract infection	19.4% [13.1–27.9]	14.8% [7.7–26.6]	0.612
Urogenital infections	12% [7.2–19.5]	18.5% [10.4–30.8]	0.381
Intra-abdominal infection	3.7% [1.4–9.1]	9.3% [4–19.9]	0.161
Skin and soft tissue infection	14.8% [9.3–22.7]	13% [6.4–24.4]	0.937
Bone and joint infection	13% [7.9–20.6]	9.3% [4–19.9]	0.666
Drug abuse	3.7% [1.4–9.1]	1.85% [0.3–9.8]	0.666
Alcohol abuse	13% [7.9–20.6]	7.4% [2.9–17.6]	0.427
Profound dental treatments	13.9% [8.6–21.7]	14.8% [7.7–26.6]	1

Data are presented as a mean or percentage. Wherever appropriate, the confidence interval was provided.

**Table 2 jcm-12-00822-t002:** Preoperative condition.

	Patients < 75 Years	Patients ≥ 75 Years	*p*-Value
Localization of the affected valve			0.789
Aortic valve	60.94%	65.57%	
Mitral valve	29.69%	29.51%	
Tricuspid valve	7.03%	4.92%	
Pulmonary valve	2.34%	0%	
Germs			0.746
*Staphylococcus aureus*	31.37%	20%	
*Staphylococcus epidermidis*	10.78%	9.09%	
Other *Staphylococci*	4.9%	3.64%	
*Enterococcus faecalis*	16.67%	21.82%	
Other *Enterococci*	0.98%	3.64%	
*Streptococcus mitis*	7.84%	9.09%	
Other *Streptococci*	14.71%	18.18%	
Other germs	12.75%	14.55%	
Multidrug resistant	0.93%	0%	1
Fever	75%	79.63%	0.646
SIRS	34.26%	46.3%	0.189
Pulmonary edema	18.52%	18.52%	1
Acute renal failure	2.78%	5.56%	0.401
Delirium	0.93%	5.56%	0.108
Cardiac arrhythmia	2.78%	7.41%	0.223
Pleural effusion	7.41%	9.26%	0.919
Heart failure	39.81%	44.44%	0.693
Septic embolism	42.59%	20.37%	**0.009**
Localization of septic embolism			**0.019**
Cerebral	38.57%	68.75%	
Spleen	24.29%	6.25%	
Kidney	15.71%	0%	
Peripheral extremities	14.29%	12.5%	
Lung	4.29%	0%	
Liver	2.86%	0%	
Diagnostic time frame	66.85 ± 49.53	50.98 ± 30.55	0.081
Preoperative laboratory findings			
Hemoglobin (mmol/L)	6.94 ± 1.38	6.58 ± 1.06	0.227
Hematocrit	0.33 ± 0.06	0.31 ± 0.05	0.244
Leukocytes (Gpt/L)	9.74 ± 4.39	10.34 ± 4.61	0.471
Platelets (Gpt/L)	243.76 ± 112.94	235.37 ± 112.47	0.605
CRP (mg/L)	83.98 ± 78.65	89.38 ± 76.51	0.425
Prothrombin time (%)	88.69 ± 23.21	78.08 ± 25.96	**0.008**
International normalized ratio (INR)	1.13 ± 0.4	1.28 ± 0.59	**0.007**
NT-proBNP (pg/mL)	6736.59 ± 10,198	10,291.6 ± 10,664	**0.002**
PCT (ng/mL)	2.37 ± 5.67	3.87 ± 10.69	0.559
Creatinine (µmol/L)	130.96 ± 118.27	134.37 ± 73.69	**0.001**
Glomerular filtration rate (mL/min)	73.91 ± 34.69	54.11 ± 24.08	**<0.001**
Albumin (g/L)	31.32 ± 7.36	31.2 ± 5.06	0.92
GOT (µkat/L)	0.82 ± 1.73	0.92 ± 1.18	0.086
GPT (µkat/L)	0.61 ± 0.85	0.72 ± 0.73	0.102

SIRS: systemic inflammatory response syndrome; CRP: C-reactive protein; NT-proBNP: N-terminal probrain natriuretic peptide; PCT: procalcitonin; GPT: glutamate–pyruvate transaminase; GPT: glutamate-oxalacetate-transaminase.

**Table 3 jcm-12-00822-t003:** Outcome and postoperative course.

	Patients < 75 Years	Patients ≥ 75 Years	*p*-Value
Postoperative laboratory findings			
Hemoglobin (mmol/L)	5.93 ± 0.9	5.88 ± 1	0.8
Hematocrit	0.28 ± 0.04	0.27 ± 0.04	0.574
Leukocytes (Gpt/L)	10.62 ± 5.69	10.92 ± 5.37	0.656
Platelets (Gpt/L)	251.82 ± 124.31	209.42 ± 113.1	0.063
CRP (mg/L)	134.3 ± 98.38	112.3 ± 78.8	0.475
Prothrombin time (%)	85.33 ± 23.68	82.1 ± 20.22	0.305
International normalized ratio (INR)	1.13 ± 0.25	1.2 ± 0.38	0.196
PCT (ng/mL)	3.15 ± 7.89	1.15 ± 3.35	0.301
Creatinine (µmol/L)	127.95 ± 82.01	130.06 ± 63.63	0.375
Glomerular filtration rate (mL/min)	71.42 ± 39.09	58.03 ± 32.55	0.123
Albumin (g/L)	24.08 ± 4.92	23.7 ± 4.89	0.837
GOT (µkat/L)	1.98 ± 0.39	1.13 ± 1.01	0.562
GPT (µkat/L)	0.53 ± 0.97	0.65 ± 0.5	0.832
Postoperative complications			
Delirium	20%	42.5%	**0.013**
CIP/CIM	4.21%	20.51%	**0.005**
Bleeding	14.74%	21.95%	0.435
Pleural effusion	29.47%	43.9%	0.151
Low-cardiac-output syndrome	4.21%	10.26%	0.23
Hospitalization cardiology	10.88 ± 9.16	16.79 ± 15.29	**0.004**
Hospitalization cardiac surgery	17.36 ± 12.64	18.38 ± 16.56	0.914
In-hospital mortality	29.91%	40.38%	0.256

CRP: C–reactive protein; PCT: procalcitonin; GPT: glutamate–pyruvate transaminase; GPT: glutamate-oxalacetate-transaminase; CIP/CIM: critical illness myopathy and critical illness polyneuropathy.

## Data Availability

Data will not be published for privacy reasons, and will be saved at the clinic.

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
