# Peer review of "Does Age Influence the Preoperative Condition and, Thus, the Outcome in Endocarditis Patients?"

_jcm, 2023, doi:10.3390/jcm12030822_

Round 1

Reviewer 1 Report

Thank you for the opportunity to review this manuscript. The authors retrospectively analyzed the situation of elderly patients (>75 years) with endocarditis and compared the preoperative and postoperative clinical status with a younger group of patients (<75 years). They concluded that although mortality was not statistically significantly higher in these older patients, perioperative complications were.

The authors provided a very concise manuscript from Introduction to Methods and Results to Discussion and Conclusion based on a straightforward research question.

However, there are several points that need further clarification:

1) How was mortality defined? Intra-hospital? Time frame? The authors should define this clearly from the beginning. With a different definition the authors might achieve a different statistical significance? Please explain/define in more detail.

2) The authors should define more precisely the role of their institution in relation to this patient database in the "Method" section. Was their institution the referring or “receiving” institution for the surgery or did they have the role of an administrator of the database? The text in the Limitation section on P7 in L 206: "... the registry's database, which includes multiple hospitals,..." raises these questions. Please explain and provide further information on this. A local study with only one centre that performed the surgery might not convey the same generalizability as a study with multiple centers.

3) There may be a bias in patient selection. According to the authors, only patients who were referred or scheduled for cardiac surgery were included in the study. Thus, there was pre selection by the physician, referring hospital or heart team. Please explain the relevant context. Thus, could it be that older patients (>75 years) with high frailty or ESII were less likely to be referred for surgery and included in the study? If not, could the outcome be much worse than in younger patients. Please explain, also in light of comment 2.

4) The authors refer to the concept of patient frailty throughout the discussion section. However, they have not provided a score or scale for it. Please clarify this.

5) On P5 in L 132 to 134: The authors state that "laboratory parameters were comparable postoperatively: ....". However, no data was shown. Please provide them, even if only as supplementary material. This would strengthen the study.

6) On P6 in L 183/184: The authors write: "Early mortality was high in both cohorts, consistent with many other recent studies." - Please include a reference for this claim and explain it in the Discussion section. This could provide a broader context for their study.

7) On P5 in L132: there are some inconsistencies: "The length of hospital stay was longer in older patients (Table 2)." but the duration is given in Table 3.

Author Response

Reviewer 1

Thank you for the opportunity to review this manuscript. The authors retrospectively analyzed the situation of elderly patients (>75 years) with endocarditis and compared the preoperative and postoperative clinical status with a younger group of patients (<75 years). They concluded that although mortality was not statistically significantly higher in these older patients, perioperative complications were.

The authors provided a very concise manuscript from Introduction to Methods and Results to Discussion and Conclusion based on a straightforward research question.

However, there are several points that need further clarification:

  • Answer: Dear Reviewer, thank you for your thoughtful reading, careful review and comments, which contributed to the improvement of the manuscript. Below we have answered the comments and marked the corresponding changes in the text in yellow.

comment 1) How was mortality defined? Intra-hospital? Time frame? The authors should define this clearly from the beginning. With a different definition the authors might achieve a different statistical significance? Please explain/define in more detail.

  • Answer 1: The question is perfectly justified. Early mortality refers to In-hospital mortality. We have now added this everywhere in the corresponding place in text.
  • Change 1: Page 1: abstract, page 2: graphical abstract, page 5: section postoperative course and complications, page 6: table 3 and discussion, page 8 conclusion.

Comment 2) The authors should define more precisely the role of their institution in relation to this patient database in the "Method" section. Was their institution the referring or “receiving” institution for the surgery or did they have the role of an administrator of the database? The text in the Limitation section on P7 in L 206: "... the registry's database, which includes multiple hospitals,..." raises these questions. Please explain and provide further information on this. A local study with only one centre that performed the surgery might not convey the same generalizability as a study with multiple centers.

  • Answer 2: Thank you for this very important comment. We have made a mistake in the limitation. This is a prospective multicenter trial. Our cardiac surgery department acts simultaneously as initiator with several cooperating hospitals. In the submission process, in the pressure to keep the deadline for the special issue, we forgot to name the authors of the cooperation departments. The additional information are now marked in yellow.
  • Change 2: page 1, page 2: section Methods, page 8: Limitations

Comment 3) There may be a bias in patient selection. According to the authors, only patients who were referred or scheduled for cardiac surgery were included in the study. Thus, there was pre selection by the physician, referring hospital or heart team. Please explain the relevant context. Thus, could it be that older patients (>75 years) with high frailty or ESII were less likely to be referred for surgery and included in the study? If not, could the outcome be much worse than in younger patients. Please explain, also in light of comment 2.

  • Answer 3: You are absolutely right. The aim of this study is mainly to analyze the outcome of endocarditis patients after cardiac surgery in association to age. In this study, only patients who underwent surgery are included. Some of the patients referred for surgery were not operable due to high risks and severe frailty. Some of them were certainly not referred for surgery for the same reason. If every patient regardless of age and frailty received surgery, the outcome would have been significantly worse for elderly patients. This is an important point for discussion, which we have now explained in the corresponding section thanks to your comment.
  • Change: Page 2: Introduction, page 3: Patients and Methods, exclusion criteria, page 7: Discussion

Comment 4) The authors refer to the concept of patient frailty throughout the discussion section. However, they have not provided a score or scale for it. Please clarify this.

  • Answer 4: That's right. The topic of frailty is still new in cardiac surgery. Although more and more cardiac surgeons are now taking frailty into account when making decisions about surgery, there is no united quick and uncomplicated method for testing of frailty in daily clinical practice. This has led to the patients being assessed very differently and unsystematically depending on the surgeon. These mainly included hand pressure, 5-minute walk test, mobility, comorbidities and corresponding symptoms and limitations as well as mental tests such as the Mini Mental State Test This heterogeneity is the reason that frailty found no place in the method or analysis and was only mentioned in the discussion. We adjusted the discussion respectively adding that frailty needs to be assessed as a compulsory addendum to the somatic EuroSCORE given the steadily increasing age and comobrbidities of our patients. For instance using the CSF
  • Change: page 6, discussion

Comment 5) On P5 in L 132 to 134: The authors state that "laboratory parameters were comparable postoperatively: ....". However, no data was shown. Please provide them, even if only as supplementary material. This would strengthen the study.

  • Answer 5: We agree. In order to keep manuscript short, we added the postoperative laboratory parameters in Table 3.
  • Change: page 5 and 6, table 3

Comment 6) On P6 in L 183/184: The authors write: "Early mortality was high in both cohorts, consistent with many other recent studies." - Please include a reference for this claim and explain it in the Discussion section. This could provide a broader context for their study.

  • Answer 6: This sentence is unfavorable worded. The phrase should mean that overall mortality in endocarditis patients is very high, regardless of age. We have now formulated it more clearly and given a source.
  • Change: page 7 and references

Comment 7) On P5 in L132: there are some inconsistencies: "The length of hospital stay was longer in older patients (Table 2)." but the duration is given in Table 3.

  • Answer 7: Thank you for your thoughtful comment. Unfortunately we made a mistake. The data for hospitalization is given in table 3. Table 2, which was incorrectly specified, has now been changed to Table 3.
  • Change: page 5: postoperative course and complications

Author Response

Reviewer 2

The authors have attempted to explain differences between more elderly (54 patients) and less elderly patients (108 patients) that underwent cardiac surgery and developed post-op infectious endocarditis. A certain criticism has to be manifested.

  • Answer: Thank you for reading our manuscript and reviewing it. We have tried to respond to your comments as good as we could. All changes are marked in yellow in the text.

Comment 1) In the abstract the authors use acronyms without any legends. That must be fixed.

  • Answer 1: this has now been corrected.
  • Change: page 1, abstract

Comment 2) The major point is that authors reported several factors that differ in the elderlies with respect to the younger people but were not significant. They report that “Proportion of patients with recurrent endocarditis, patients with early endocarditis (within 12 months after prosthesis implantation), proportion of redo surgery, electrophysiology devices such as pacemakers or defibrillators, patients with carcinoma or recent  chemotherapy, intra-abdominal and urogenital infections, and dental treatments showed no significant difference between the age groups, although the above items were more frequent in the elderly”.   And for this reason those variables were substantially excluded in explaining the outcome. However the authors could have incurred in a major statistical problems: the type 2 errors that means that they exclude a statistical difference only because the study is underpowered but this difference does exist. This can happen exquisitely for the small sample size. In fact the power of a testy is very dependent on the size of the sample. Ideally we would want an 80% chance of detecting a relationship (if in fact one does exist). If you obtain a non-significant result and are using quite a small size (like the small elderly group), you could check the power value of the statistical test. Obviously the best way to proceed is to calculate a sample size of a study on the basis of a hypothesized major end-point difference (like mortality). I have included a computation of the sample size required to detect a significant difference of abdominal infection complication based on the difference reported in  the paper (Table 1) . As you can see the sample size (in bold)  is much larger than that in the study. This check of the statistical power must be done and reported and eventually the sample size adjusted. 

  • Answer: At the current point in time, the study only allows access to this so far limited patient collective, which makes increasing the sample size a question for the future. While we agree, that the mentioned differences in baseline characteristics might reach significance in a largely increased dataset, we deemed it inappropriate to discuss characteristics without significant differences as confounders at the current stage. To underline this we have calculated the 95% confidence intervals for the debated baseline characteristics (see table 1), which showed to almost always (except for Intra-abdominal infection) completely overlap. In other words: at the current stage we can not claim knowledge about a higher rate of for example early endocarditis in elderly patients and therefore should not discuss this as an influencing factor. While we agree that one could perform and report sample size calculations for all non-significant baseline characteristics, we doubt that this is appropriate for the submitted paper, as it would add a lot of data with very little relation to the discussed topic. Instead we added the following line into the discussion at line 239-240: “A future larger dataset might also be sufficient to study the effect of those baseline characteristics that were different between age groups without reaching significance. “ Regarding the general point of not excluding differences in baseline characteristics as possible influence on the outcome we would also like to mention that under this assumption no baseline characteristic can ever be ruled out. If for example Group A had an Endocarditisrate of 10.00 and Group B one of 10.01, one would then have to argue that the sample size is just too small as a sample size of several 10s of thousands would likely lead to a significant difference between the two.
  • Change: page 8, Limitations and table 1

Comment 3) I think that any kind of implants is a possible nest for bacteria. That has been quite recently explained in particular regarding pacemaker lead implant [1].

  • Answer 3: we agree, because we did not investigate this point in detail, we mentioned it in passing. g. in discussion: “In view of the significant increase in patients who have received electrophysiological implants or valve prostheses, who are chronically dialysis-dependent or who take immunosuppressants, an increased susceptibility to infections per se and thus to a susceptibility to device infections, but also to infections of the native valves, is to be expected.” Or e.g. in Introduction: “Diverse reasons contribute to this, a suppressed immune system, a cancer disease, more frequent treatments as outpatient or inpatient, dialysis dependency, valve prosthesis carrier, interventions and surgical procedures are some of the reasons that can lead to infection”. Now we have highlighted it extra in discussion.
  • Change: page 7, Discussion and reference 17

Comment 4) Hidden root canal infection is more than an issue in this context. That must be covered more in depth.

  • Answer 4: Root abscess and infection and dental treatments are indeed an important and serious problem, even if in the last guidelines have been treated very liberally. In discussion we have now expanded the topic a bit.
  • Change 4: page 7, discussion

Comment 5) Another major point is the ischemic dementia that is more than an issue after the cardiac surgery in extracorporeal circulation. That must be covered too.

  • Answer: Ischemic dementia was not the focus of this study. In view of the current evidence that around 5 to 10% of the aging population suffer from it according to a recent Canadian survey (https://www.canada.ca/en/public-health/services/publications/diseases-conditions/dementia-highlights-canadian-chronic-disease-surveillance.html) a similar proportion of our patients may have suffered from it influencing morbidity and mortality. We did not systematically aimed to identify first signs of dementia but we certainly treated only those patients who were able to voluntarily consent to surgery or other conservative or interventional treatment. Thus, patients with progressive dementia were most likely not among our cohort. This is certainly a highly interesting topic for future studies and we will implement adequate tests to elucidate that. We added a respective paragraph to the discussion.
  • Change: page 7, discussion

Round 2

Reviewer 1 Report

Thank you very much for the opportunity to review this manuscript.

The authors have answered all my comments appropriately.

Reviewer 2 Report

tha paper has been improved.